# New Species and New Records of *Strumigenys* Smith, 1860 (Formicidae: Myrmicinae) from the Neotropical *schulzi* Species Group †

**Esperidião Alves dos Santos-Neto** [1] , **Júlio Cezar Mário Chaul** [2] and **Jacques Hubert Charles Delabie** [1,3,*]

[1] Programa de Pós-Graduação em Zoologia, Universidade Estadual de Santa Cruz, Ilhéus 45662-900, BA, Brazil; easneto@uesc.br

[2] Departamento de Biologia Geral, Universidade Federal de Viçosa, Viçosa 36570-900, MG, Brazil; julio.chaul@ufv.br

[3] Comissão Executiva do Plano da Lavoura Cacaueira, Centro de Pesquisas do Cacau, Laboratório de Mirmecologia, Ilhéus 45600-000, BA, Brazil

* Correspondence: jacques.delabie@agro.gov.br

† urn:lsid:zoobank.org:act:22EB3FC1-B94B-4505-84E8-2C1886D8A9A6; urn:lsid:zoobank.org:pub:217E590F-5F81-4609-814F-11E08EB6FFC1.

**Abstract:** The genus *Strumigenys* is the third most diverse among ants, having been reviewed globally. Despite this comprehensive review, new species are frequently discovered in most biogeographic regions. Here, we describe two new species, *Strumigenys itannae* sp. nov. and *Strumigenys xoko* sp. nov., based on material collected in the Amazon and the Brazilian Atlantic Forest. Additionally, new records for *S. castanea*, *S. metrix*, and *S. orchibia* are provided, expanding the known distribution of these species. We discuss morphological variation in *S. schulzi* and provide biological notes that indicate some species within the *schulzi* group are arboreal inhabitants. We provided an identification key for the newly described species as an amendment to the identification key for the Neotropical *Strumigenys*.

**Keywords:** Attini; arboreal ants; leaf-litter ants; Amazon; Brazilian Atlantic Forest





## 1. Introduction

*Strumigenys* Smith, 1860 is a hyperdiverse ant genus that currently comprises 879 extant species globally [1]. The implementation of new methodologies for studying soil and leaf-litter fauna [2,3] has progressively increased the known diversity of *Strumigenys*, and several species are described worldwide each year [4–10]. Recent molecular evidence supports the hypothesis that the genus is monophyletic with a complex evolutionary history, which explains its division into several different genera by some authors in the past [11]. The genus is characterized by significant morphological diversity, particularly in its mandibles, which include gripping mandibles, short-trap jaws and long-trap jaws, with repeated evolution of the latter two in the different biogeographical regions, which explains minor differences observed between the same forms on several continents (e.g., short trap jaws of the Afrotropics versus in the Neotropics) [11].

The Neotropical *schulzi* group of species is a grouping defined through morphological similarities and geographic distribution [12] and is characterized by its short, gripping mandibles. Molecular evidence recovered the group as paraphyletic, with its species divided into two distinct clades: one composed of species related to *Strumigenys schulzi* and the other of *Strumigenys microthrix* (Kempf, 1975) [11]. Currently, the *schulzi* group comprises 20 species, and since the revision by Bolton [4], three species have been described for the group: *Strumigenys aequinoctialis* DeAndrade, 2007 [5], *Strumigenys subnuda* MacGown & Hill, 2010 [13] and *Strumigenys madrigalae* Latkke and Aguirre, 2015 [7].

In the past 25 years, due to the development of ant studies in Brazil, many specimens of *Strumigenys*, including some of the *schulzi* group, have accumulated in the country's

myrmecological collections without receiving special attention. This study describes two new species and adds new recorded species to the Neotropics. We also discuss observed morphological variation and provide notes on the biology of some species.

## 2. Materials and Methods

*Repository institutions.* Formicidae Collection at Centro de Pesquisas do Cacau, Ilhéus, Bahia, Brazil (CPDC) [14]; Instituto Nacional de Pesquisas da Amazônia, Coleção Sistemática da Entomologia, Manaus, Amazonas, Brazil (INPA); and Coleção Entomológica do Laboratório de Coleoptera, at the Universidade Federal de Viçosa, Minas Gerais, Brazil (CELC, collection acronym as Evenhuis) [15].

*Morphological data set.* We primarily follow Bolton's terminology [4], except for mesosoma instead of "alitrunk" [16]. We followed [17,18] for body measurements and their abbreviations.

*TL*. Total Length: A proxy for the total length of the body, given by the sum of ML, HL, WL, PetL + A3 + A4L.

*HL*. Head Length: measured from the midpoint of the occipital margin to the midpoint of the anterior clypeal margin.

*HW*. Head Width: measured at the maximum width of the head in full-face view.

*ML*. Mandible Length: measured from the mandibular apex to the anterior clypeal margin.

*SL*. Scape Length: measured from the basal constriction that occurs distal of the condylar bulb.

*EL*. Eye Length: maximum diameter of the eye.

*PrW*. Pronotal Width: maximum width of the pronotum in dorsal view.

*WL*. Weber's Length: diagonal length of the mesosoma in profile view.

*PetL*. Petiolar Length: maximum length of petiole.

*PetH*. Maximum distance between the apex and the most ventral point of the petiolar node.

DPetW

*DPetndL*. The maximum length of the petiole node in dorsal view of the metasoma

*PosPetW*. The maximum width of the postpetiole disc in dorsal view.

*A3 + A4L*. The distance from the anterior-most point of the postpetiole (spongiform collar not considered) to the posterior-most point of the first gastral tergite (Figure 1C).

*GW*. The maximum width of first gastral tergite in dorsal view of the metasoma (A4).

*MtbtL*. Metabasitarsus maximum length.

*Mtfml*. Metafemur length: maximum length of the metafemur, not including the trochanter.

*MttbL*. Metatibia length: maximum length of the metatibia.

*MtbtL*. Metabasitarsus maximum length.

*Imaging*. A Leica S8APO stereomicroscope coupled with a Canon 1100D or a Leica M165C stereomicroscope coupled with a Leica DMC 290 were used to acquire the images. We used the dome model presented in [19] for the illumination. Acquired images were stacked in the Zerene stacker software 1.04 [20] and Helicon Focus 8.2.2 pro [21]. Images were edited to enhance sharpness, adjust rotation and light intensity in Gimp [22]. Scales bars are made with the software ImageJ [23] by calibrating the program with a measurement taken during the acquisition of the images. The map was made using QGIS software [24].

*Species delimitation method*. We compared the sets of morphological characters of the new species within the genus *Strumigenys*, which allowed us to comfortably classify them as new species. The discontinuity observed in these characters served as an indirect indication of the reproductive isolation of the species described here from the known species of the genus.

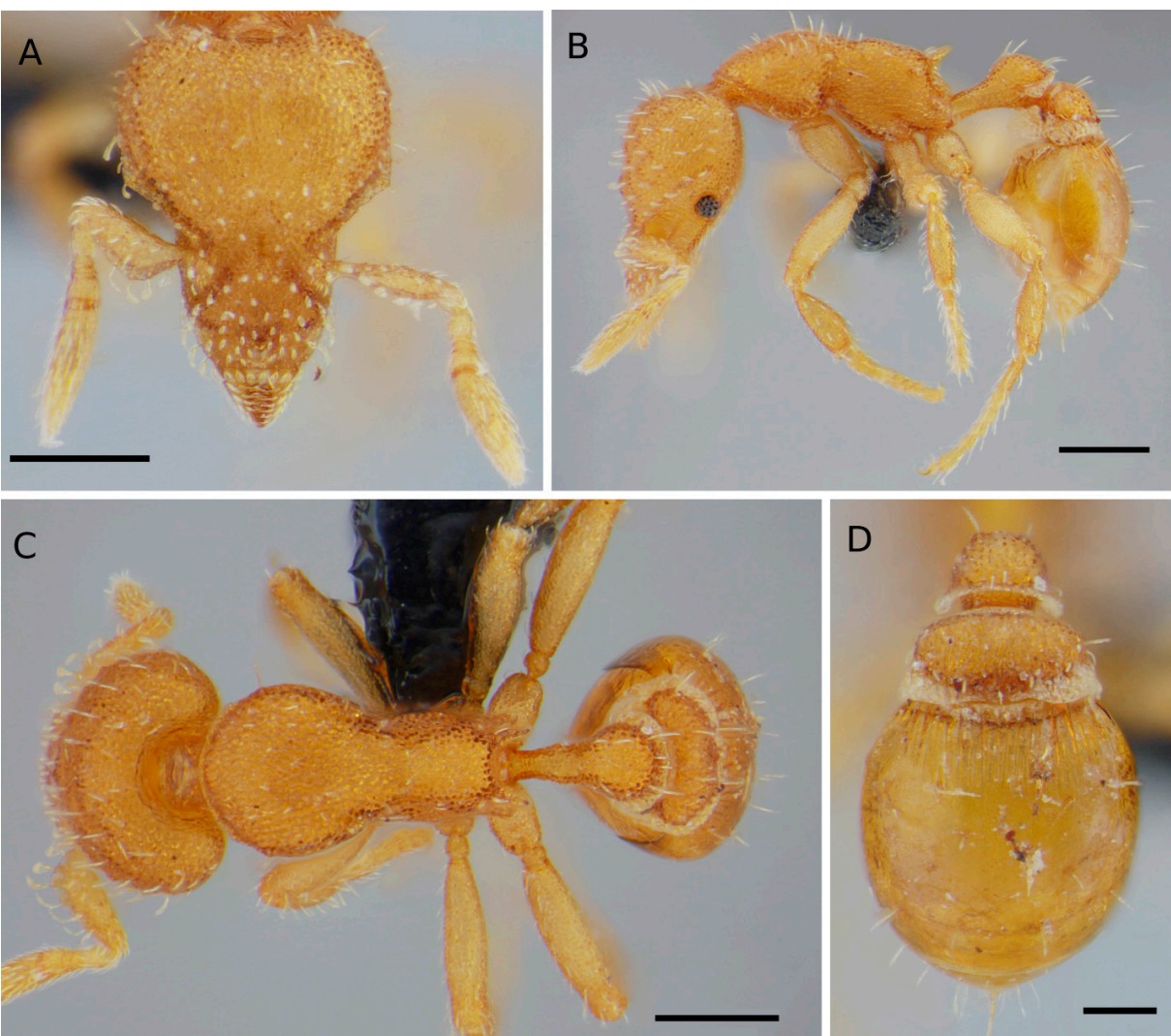

**Figure 1.** *Strumigenys itannae* sp. nov., holotype worker, head in full-face view (**A**), lateral view (**B**), dorsal view (**C**), gaster in dorsal view with basigastral costulae in detail. (**D**). [ANTWEB1032113]. Scale bars are 0.2 mm.

## 3. Results

*New records* Strumigenys schulzi *species group*
*Strumigenys castanea* (Brown, 1953) [12]
**BRAZIL**, **Bahia**, Santo Amaro, 11.xii.1987, (Ferraz, E.C) [one worker].
*Strumigenys emiliae* Forel, 1907 [25]
**BRAZIL, Bahia,** Buerarema, −14°45′30″, −39°13′88″, 18.iv.1998, (Santos, J.R.M. dos) [three workers]. Camamu, −13°58′00″, −39°15′33″, 01.v.1992, (Eduardo, J.) [one worker]. Canavieiras, −15°32′07″, −39°00′31″, 20.vi.1998, (Santos, J.R.M. dos) [two workers]. −15°40′85″, −39°00′26″, 17.vii.1998, (J.C.S. Carmo & J.R.M. Santos) [three workers]. Ibirapitanga, −14°11′39″, −39°25′23″, 23.x.1980, (Forbes Benton) [one worker]. Iguaí, −14°38′35″, −40°08′52″, 23.x.1980, (C. Leite & T. Porto et al.) [two workers]. Ilhéus, −14°45′21.2″, −39°04′14.7″, 16.ix.1993, (J.E. Silveira) [one worker]. −14°46′35.1″, −39°13′51.4″, 19.ii.1998, (J.D. Majer) [one worker]. −14°46′16.6″, −39°13′50.3″, 14.ix.1993, (R.M.F. Cordeiro) [four workers]. −14°46′13.3″, −39°13′28.9″, 28.iv.1996, (J.D. Majer) [one worker]. −14°46′35.1″, −39°13′51.4″, 15.iv.1997, (J.R. Maia) [one worker]. Itabuna, −14°46′41″, −39°17′40″, 19.i.1998, (Santos, J.R.M. dos) [one worker]. Ubaíra, −13°07′18″, −39°40′13″, 03.vii.1905, (C. Leite & T. Porto et al.) [two workers]. Wenceslau Guimaraes, −13°33′34″, −39°42′07″,

06.vii.2014, (C. Leite & T. Porto et al.) [one worker]. **Rio Grande do Sul**, Santa Maria, −29°39′35.2″, −53°54′37.3″, 2011/2012, (Jadel Boscardin) [one worker].

Strumigenys epinotalis Weber, 1934 [26]

**BRAZIL, Bahia**, Ilhéus, −14°45′55.5″, −39°13′55.1″, 11.xi.1992, (Ivan) [two workers]. Barra do Choça, −14°49′27.76″, −40°38′49.28″, 20.i.2011, (J. Martins-Freitas) [one worker]. −14°48′55.68″, −40°38′46.19″, 20.i.2011, (J. Martins-Freitas) [one worker]. Itamaraju, −17°01′41.0″, −39°33′06.4″, 15.iv.1981, (Forti) [one worker]. **FRENCH GUIANA**, Saint-Elie, 4°48′08.3″, −53°16′51.5″, abr/02, (J. Orrivel & J. Le Breton) [seven workers].

*Strumigenys grytava* (Bolton, 2000) [4]

**BRAZIL, Bahia**, Camaçari, −12°43′05.7″, −38°19′47.3″, 08.viii.2012, (Melo, T.S.) [three workers]. Canavieiras, −14°24′34″, −39°01′00″, 30.iii.1998, (J.C.S. Carmo & J.R.M. Santos) [one worker]. Conde, Barra do Itariri, −11°56′58″, −37°36′34″, 06.x.2010, (Travassos, M.L.O.) [one worker]. Cruz das Almas, −12°41′03.9″, −39°06′32.8″, 18.i.1995, (M.R.B. Smith) [one worker]. Ilhéus, Barramares, −14°36′30.6″, −39°03′38.0″, 22.IV.2010, (José, R.M. dos Santos) [one worker]. Olivença, −14°54′57″, −39°03′63″, 20.xi.1998, (Santos, J.R.M dos) [one worker]. Ponta do Ramo, −14°29′37″, −39°02′15″, 02.x.1998, (Carmo, J.C.S.) [one worker]. Itacaré, Faz. Sempre Viva, −14°21′59.7″, −39°02′08.3″, 28.v.1991, (Ismael Rosa) [two workers]. Pasto, −14°18′52″, −39°02′10″, 12.x.2010, (J.R.M, J.C & J.A.) [three workers]. Itubera, Reserva Michelin, −13°51′01″, −39°13′34″, 25.vii.2008, (Leg. Resende, J.J. Silva, E.M, Gomes, A.R.) [one worker]. Igrapiúna, Reserva Michelin, −13°49′19.2″, −39°10′02.7″, 05.ix.2012, (Varjão, S.L.S.) [one worker]. Lauro de Freitas, Busca Vida, −12°51′51″, −38°16′11″, 06.x.2010, (Travassos, M.L.O.) [one worker]. Marau, −14°12′65″, −39°03′48″, 04.xii.1998, (J.C.S. Carmo & J.R.M. Santos) [three workers]. Mucugê, Meio montane, −13°00′27.7″, −41°21′50.0″, 6-12.xii.1990, (Brandão, Diniz e Oliveira) [five workers]. Salvador, −12°56′06.8″, −38°24′47.9″, 12.ix.2012, (Melo, T.S.) [two workers]. Ubaíra, −13°07′18″, −39°40′13″, 03.vii.1905, (C. Leite, T. Porto et al.) [two workers]. Uruçuca, Faz. Sempre Viva, −14°21′44″, −39°02′08.9″, 28.v.1992, (Ismael Rosa) [12 workers]. Serra Grande, −14°27′05″, −39°02′34″, 28.iv.1997, (Carmo, J.C.S.) [one worker]. −14°26′48″, −39°02′56″, 02.v.1998, (Carmo, J.C.S.) [four workers]. Valenca, −13°19′51″, −39°11′27″, 2011, (C. Leite, T. Porto et al.) [two workers]. Wenceslau Guimarães, −13°33′34″, −39°42′07″, 03.vii.1905, (C. Leite, T. Porto et al.) [three workers]. −13°33′34″, −39°42′07″, 03.vii.1905, (C. Leite, T. Porto et al.) [one worker].

*Strumigenys margaritae* Forel, 1893 [25]

**FRENCH GUIANA**, Sinnamary, Paracou Station, −05°17′213″, −52°55′261″, 13.iv.2009, (Sarah Groc et al.) [fifteen workers]. 05°17′, −52°54′, 12.ii.2010, (Sarah Groc et al.) [two workers]. Paracou Station, −05°17′213″, −52°55′261″, 23.iv.2009, (Sarah Groc et al.) [one worker]. **F.W.I. MARTINIQUE**, Chemim Elisabeth, 14°43′19.1″, −61°00′40.9″, 27.ii.2014, (Leg. S. Quinquenel) [one worker].

*Strumigenys metrix* (Bolton, 2000) [4]

**BRAZIL, Bahia**, Ilhéus, Cepec-F, −14°45′34″, −39°13′45″, 06.viii.92, (M.R.B. Smith), 4546, [one worker]. **Rondônia,** Ouro Preto do Oeste, CEPLAC, 10°43′01.47″ S, 62°13′48.86 W.

*Strumigenys urrhobia* (Bolton, 2000) [4]

**BRAZIL, Amazonas**, Manaus, −2°55′52.8″, −59°57′14.2″, 06.vi.1991, (F.P. Benton) [one worker]. Bahia, Aurelino Leal, −14°22′58″, −39°24′56″, 26.v.1997, (Santos, J.R.M. dos) [one worker]. Arataca, −15°15′90″, −39°16′01″, 23.xi.1999, (Santos, J.R.M. dos) [four workers]. Buerarema, −14°45′30″, −39°13′88″, 18.iv.1998, (Santos, J.R.M. dos) [one worker]. −14°54′07″, −39°17′12″, 28.x.2002, (Santos, J.R.M. dos) [three workers]. Camacan, −15°25′24″, −39°25′59″, 27.vii.1993, (José Eduardo) [one worker]. −15°25′30″, −39°27′19″, 27.v.1999, (Santos, J.R.M. dos) [one worker]. −15°25′45″, −39°25′18″, 30.viii.1997, (T. Carvalho) [one worker]. −15°26′01″, −39°23′57″, 17.ix.1997, (E. Santana) [one worker]. Canavieiras, −15°10′14″, −39°18′04″, 21.vi.1997, (F. Lima) [two workers]. −15°32′04″, −39°00′39″, 14.xii.1997, (J.C.S. Carmo & J.R.M. Santos) [two workers]. Itacaré, −14°18′16″, −39°06′40″, 06.ix.1998, (E. Santos) [two workers]. −14°19′04″, −39°04′19″, 03.viii.1998, (Santos, J.R.M. dos) [six workers]. Itajuípe, −14°42′12″, −39°29′53″, 18.vi.1997, (San-

tos, J.R.M. dos) [six workers]. Iguaí, −14°38′35″, −40°08′52″, (C. Leite & T. Porto et al.) [14 workers]. Ilhéus, −14°31′46″, −39°03′34″, 28.viii.1998, (J.R. Maia) [four workers]. −14°32′72″, −39°25′39″, 06.x.1997, (J.C.S. Carmo & J.R.M. Santos) [two workers]. −14°32′17″, −39°05′12″, 10.i.1999, (M. Pereira) [two workers]. −14°33′34″, −39°00′34″, 28.viii.1998, (J.R. Maia) [four workers]. 14°36′34″, −39°16′00″, 23.vii.1997, (J.E. Silveira) [three workers]. −14°38′12″, −39°09′12″, 14.viii.2002, (T. Souza) [two workers]. −14°39′58.4″, −39°04′52.5″, 03.iii.1999, (C. Oliveira) [one worker]. −14°40′51″, −39°15′24″, 12.i.1998, (C. Leite & T. Porto et al.) [three workers]. −14°44′34″, −39°11′27″, 28.iv.1999, (A. Pereira) [two workers]. −14°45′34″, −39°13′45″, 22.iii.1991, (B. Santos) [two workers]. −14°46′34.4″, −39°13′48.2″, 24.ix.1996, (B.C. Antônio) [two workers]. −14°46′34.4″, −39°13′48.2″, 08.x.2017, (E. Carvalho & J.E. Queiroz Jr) [two workers]. −14°47′50″, −39°03′82″, 12.iv.1991, (J.E. Silveira) [two workers]. −14°47′44″, −39°10′51″, 12.iii.2003, (A. Silva) [one worker]. −14°48′86″, −39°06′00″, 03.ix.1997, (Santos, J.R.M. dos) [one worker]. −14°50′22″, −39°07′54″, 18.xi.2000, (B. Santos) [one worker]. Itabuna, −14°47′53″, −39°13′56″, 05.ix.2000, (T. Santos) [one worker]. −14°48′30″, −39°14′00″, 28.v.1998, (P. Costa) [three workers]. −14°48′50″, −39°15′05″, 16.vii.1996, (A. Almeida) [one worker]. −14°49′08″, −39°07′35″, 02.vii.1999, (C. Gomes) [two workers]. −14°49′40″, −39°08′02″, 29.iii.1999, (D. Almeida) [three workers]. Jussari, −15°08′26″, −39°31′29″, 26.v.1999, (J.C.S. Carmo & J.R.M. Santos) [five workers]. −15°11′44″, −39°26′46″, 18.vii.1999, (Santos, J.R.M. dos) [one worker]. −15°25′30″, −39°27′19″, 27.v.1999, (Santos, J.R.M. dos) [one worker]. Maraú, −14°08′22″, −39°07′53″, 16.iii.1998, (M. Lima) [one worker]. −14°09′41″, −39°09′23″, 20.vii.1996, (P. Silva) [two workers]. −14°09′45″, −39°00′40″, 30.vi.1997, (Santos, J.R.M. dos) [three workers]. Mascote, −15°34′38″, −39°24′36″, 19.iii.1999, (Santos, J.R.M. dos) [one worker]. −15°44′04″, −39°23′04″, 11.xi.1999, (Santos, J.R.M. dos) [three workers]. Porto Seguro, −16°54′31.1″, −39°13′05.5″, 17.viii.1996, (S. Silva) [two workers]. Santa Luzia, −15°26′32″, −39°22′45″, 11.vi.1998, (Santos, J.R.M. dos) [one worker]. Uruçuca, −14°31′52″, −39°07′45″, 29.ix.1996, (J. Almeida) [one worker]. −14°33′02″, −39°05′44″, 12.iv.2001, (J. Lima) [one worker]. −14°29′55″, −39°02′34″, 27.x.1999, (L. Oliveira) [two workers]. −14°24′20″, −39°05′35″, 21.vii.1996, (B. Santos) [three workers]. Una, −15°15′42″, −39°09′12″, 12.vi.1997, (Santos, J.R.M. dos) [one worker]. Valença, −13°19′11″, −39°03′59″, 21.x.1998, (F. Souza) [two workers]. −13°19′51″, −39°11′27″, (C. Leite & T. Porto et al.) [two workers]. −13°21′32″, −39°00′34″, 23.vii.1996, (A. Oliveira) [two workers]. −13°21′50″, −39°12′40″, 17.x.1998, (S. Silva) [two workers]. −13°22′40″, −39°09′30″, 16.viii.1997, (J. Santos) [three workers]. **Maranhão,** São Luís, −02°36′15.35″, −44°15′2.09″, 12.ix.2014, (Joudelys A. Silva) [one worker]. **Pará**, Goianésia, (Santos, J.R.M. dos) [one worker]. Marituba, −01°22′, −48°20′, (Santos, J.R.M. dos) [12 workers].

**FRENCH GUIANA**, Maripasoula, 3°43′21.5″, −54°00′37.1″, 01.vii.1999, (A. Dejean) [four workers].

New species described here do not modify the *Strumigenys schulzi* species group diagnosis as available in [4].

*Strumigenys itannae* sp. nov. Santos-Neto, Chaul and Delabie

(Figure 1)

*Type material.* Holotype worker: **PERU. Madre de Dios**: Reserva Nacional Tambopata, Sachavacayoc, −12.855896 −69.361939, 19–31.vii.2012, Winkler trap (Feitosa, R.; Probst, R.; Camacho, G.; Chaul, J.) [CELC, ANTWEB1032113]. Paratype worker. **BRAZIL. Rondônia**: Jaci-Paraná, −9.451220 −64.376030, 01–27.i.2013, Winkler trap (Fernandes, I.) [INPA, ANTWEB1032003].

*Non-type material.* One worker with the head missing; sample data as holotype [CELC, UFV-LABECOL-001540].

*Diagnosis.* Basal lamella of mandible triangular, without diastema between it and masticatory margin; head dorsum with several erect hairs that gradually change from smaller, suberect and spatulate; eyes large, with 18–20 ommatidia in total; and stand pilosity simple, stiff, without ground pilosity contrasting to them. Mesosoma, petiole and

postpetiole disc entirely and deeply reticulate-punctate; gaster first tergite smooth, except for basigastral costulae.

*Geographical Distribution.* Brazil (Rondônia); Peru (Madre de Dios). (Figure 2)

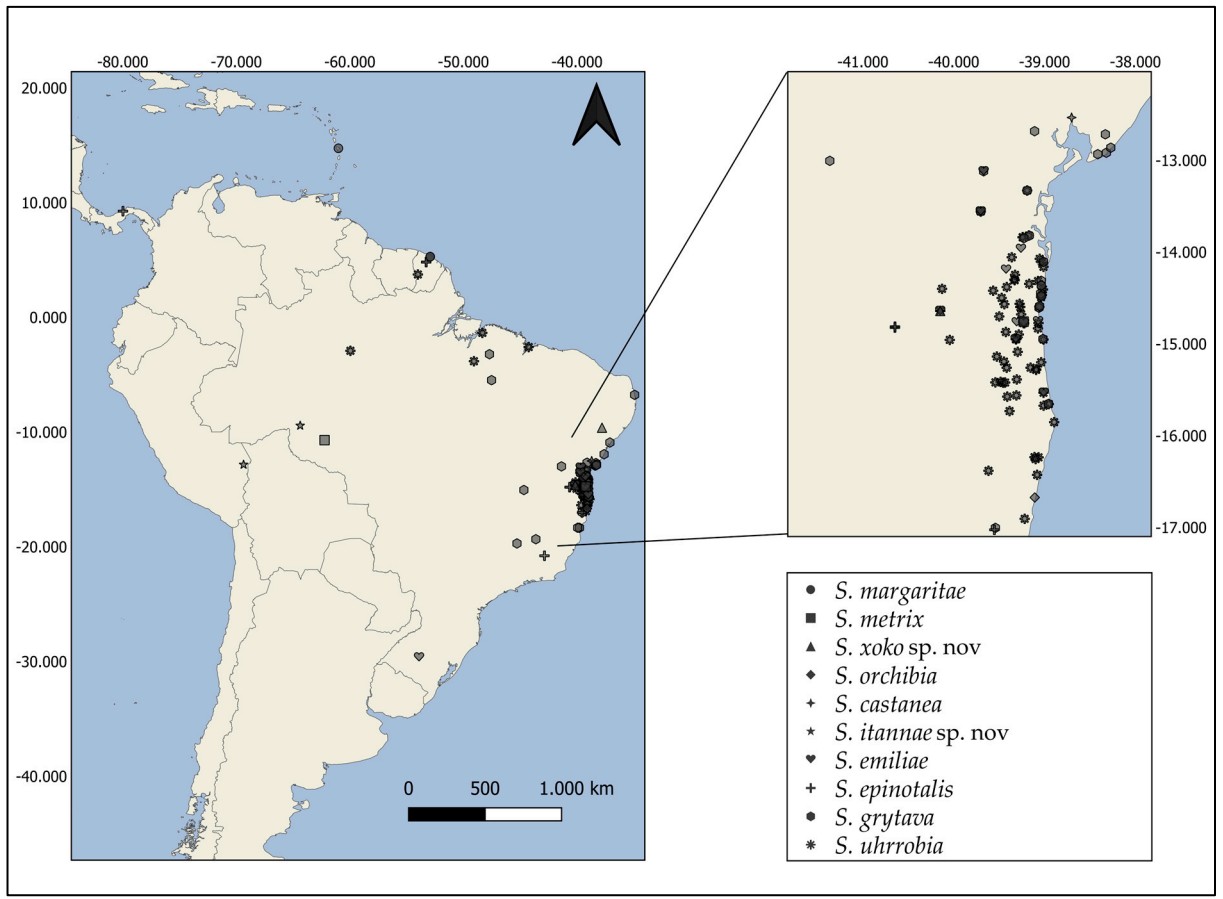

**Figure 2.** Distribution of the new species and new records for previously described species of the *schulzi* group.

*Description. Holotype.* HW 0.39, ML 0.08, ML2 0.1, HL 0.51, SL 0.21, OL 0.06, ON 20, WL 0.53, PetL 0.25, A3 + A4L 0.48, PrW 0.28, DPetW 0.13, DPetndL 0.11, PosPetW 0.22, GW 0.36, MtfmL 0.35, MttbL 0.25, MtbtL 0.18, CI 81, SI 53, MI 20, TL 1.88. *Paratype.* HW 0.39, ML 0.08, ML2 0.1, HL 0.48–0.51, SL 0.21, OL 0.06, ON 18, WL 0.51, PetL 0.26, A3 + A4L 0.46, PrW 0.28, DPetW 0.13, DPetndL 0.10, PosPetW 0.22, GW 0.36, MtfmL 0.33, MttbL 0.24, MtbtL 0.17, CI 77, SI 53, MI 20, TL 1.81. *Head.* Basal lamella of mandible triangular followed by a total of 12 teeth; basal seven more developed than the apical five; teeth one and two somewhat blunt, the first being more than the latter; tooth three acute and the largest; teeth four to seven subequal and similar in shape to two. Apical five teeth divided into four tiny denticles, and the apical one, which is slightly more developed. Dorsal surface of mandibles covered with tiny, flattened, appressed hairs. Basiventral mandible gland visible (Figure 1A). Clypeal anterior margin is overall convex, the medial section flat with freely projecting spoon-shaped hairs that are more flattened laterally; the posterior clypeal margin (epistomal sulcus) with an arch of suberect hairs, other hairs on the dorsum of clypeus appressed, spoon-shaped and smaller. Clypeus surface slightly raised in relation to the head surface posterad. Head dorsum with several erect hairs that gradually change from smaller, suberect and spatulate anteriorly to longer, thinner and erect posteriorly (Figure 1B). Eyes well-developed, with 18 to 20 ommatidia. Scrobe well delimited and shallowly concave. Preocular carina well-developed and reaches the level of the eye. In full-face view, lateral margins of the head (dorsal scrobe margins) abruptly diverge posteriorly to the antennal

insertions, the margins shelf-like, conspicuous translucent with backlighting. Antennae 6-segmented. Scape flattened, anterior margin with a well-developed sub-basal curvature and a row of freely projecting setae that vary from spoon-shaped basally to thinner apically, the first one or two proximal to the sub-basal bent are slightly curved apically, the four distal to these two curves to the base of scape and the distal two or three curved to the apex of the scape. Scape apicoventral gland visible. The head and dorsal surface of the scapes reticulate. *Mesosoma*. Entirely deeply reticulate-punctate. In profile, dorsal outline of promesonotum convex, metanotal impression poorly developed, dorsum of propodeum roughly straight; propodeal spine long and posteriorly and slightly dorsally directed; and propodeal lamella as thick as or slightly thicker than the propodeal spiracle diameter, without any lobe or indentation on its lower portion. Dorsally on mesosoma, various, simple, suberect to erect setae, of which the humeral pair and two pairs of lateral mesonotal setae are slightly longer than the remaining ones (Figure 1C). Mesopleural excavations without a conspicuous patch of small setae. Legs with suberect, slightly curved, simple to slightly remiform setae. Femora, especially meso- and metafemora, robust. Anteroproximal surface of procoxa shallowly concave, with somewhat longer setae. Protibial apical gland visible, as well as patches similar on the basitarsi of all the legs. *Metasoma*. In dorsal view, the petiole node and postpetiole disc reticulate-punctate, latter with shallow, superimposed rugulae; petiole node slightly broader than long; and postpetiole disc more than twice as large as long (Figure 1C,D). Spongiform tissue absent ventrally on the petiole, present as a thin, transversal lamella on posterior edge of petiole node, thicker laterally than medially; ventrally on postpetiole, two thick ventrolateral triangular outgrowths are linked medially by a thinner strip (Figure 1C); dorsally on postpetiole, an anterior thin lamella and a posterior lamella that is medially thin and posterolaterally enlarged; a transversal lamella on the anterior edge of first gaster tergite. Specialized basigastral ventral setae arranged as a thin transversal band, hyphae-like. Gaster smooth, except for well-marked basigastral costulae, which are slightly smaller than postpetiole disc length (Figure 1D). Erect, simple setae scattered on the dorsum of the metasoma.

*Etymology*. The specific epithet honours the Brazilian myrmecologist Itanna Oliveira Fernandes from the Instituto Nacional de Pesquisas da Amazônia–INPA.

*Biology*. The species has been sampled in the leaf litter by Winkler extractors twice in localities separated by almost 700 km in the catchment of the Madeira River.

*Comments: Strumigenys itannae* slightly resembles *S. schulzi*, *S. castanea*, *S. metrix* and *S. orchibia* within the *schulzi* group, but it can be easily isolated by the quantity of erect hairs on the vertex and mesosoma. The shape of the petiolar node is similar to that of *S. schulzi* and *S. castanea*, but it can be distinguished from these two additionally by the larger propodeal spine size and thinner, observed from the reduced propodeal lamella (Table 1).

*Strumigenys orchibia* (Brown, 1953) [12]

*New records*. **BRAZIL, Bahia**, Una, Oiticica, −16°40′35″, −39°06′26″, 01.v.2009, (J.R.M., J.C.J.A.) [one worker] [5649]. Unacau, −15°05′21″, −39°17′42″, 11.ii.2000, (Santos, J.R. dos) [one worker].

*Comments*. We recorded *Strumigenys. orchibia* [12] for the first time outside the Amazon region, in the city of Una, Bahia, Brazil, in Atlantic Forest biome (Figure 3A,B). The Una population shows some differences from the holotype locality (Figure 3C,D), namely smaller total size, lower mesopleuron with faint reticulation and small smooth areas (rather than strongly reticulate), basigastral costulae occupying less than a quarter of the first gastral tergite, dorsum of head with less dense ground pilosity. The pilosity of specimens from both regions is highly comparable, with *S. orchibia* exhibiting lower abundance of pilosity on the first gastral tergite. In the specimen from the Atlantic Forest, this trait is also evident in the ground pilosity of the head. The scarcity of specimens poses a challenge on whether these two forms represent merely geographic variations of *S. orchibia* or distinct species.

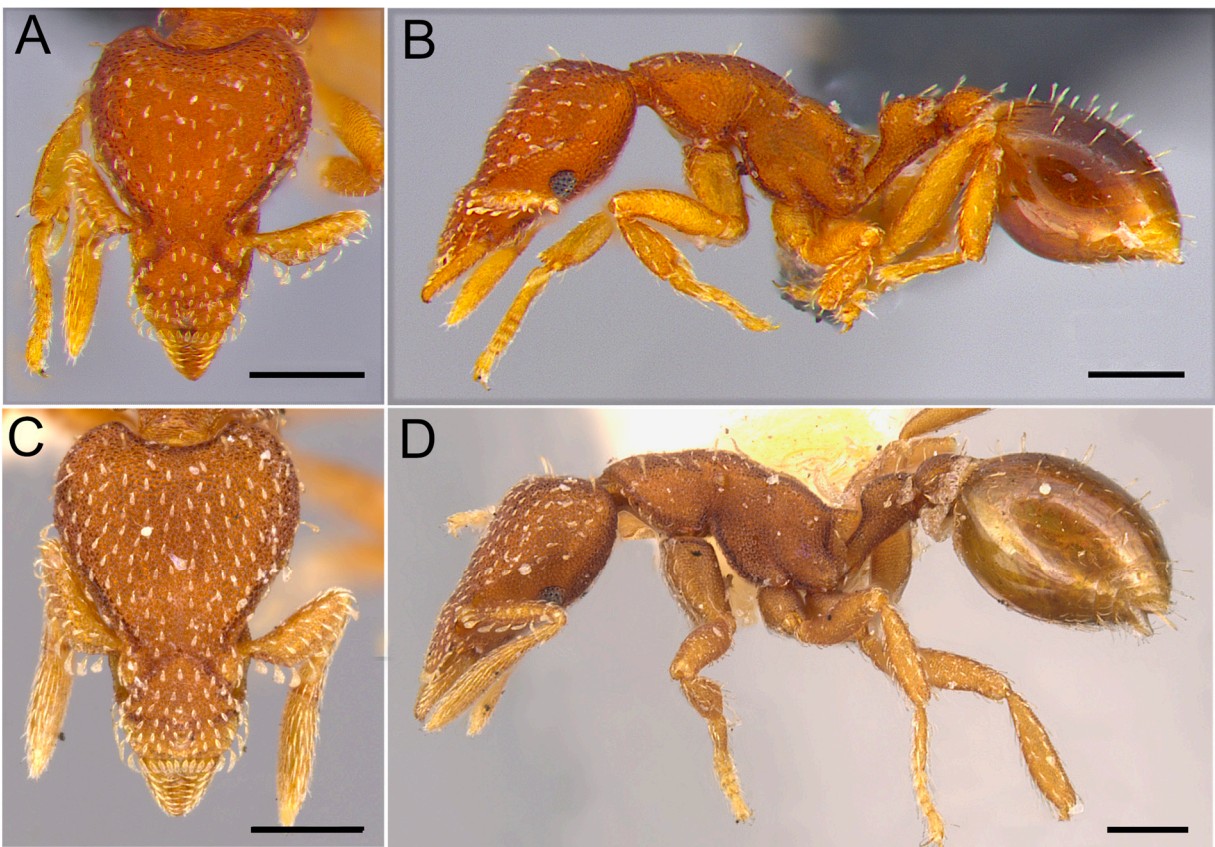

**Figure 3.** *Strumigenys orchibia*, specimen collected in Atlantic Forest, head in full face view (**A**), lateral view (**B**). Paratype [BMNH1013599]. from the Amazon region, head in full-face view (**C**), lateral view (**D**). Scale bars are 0.2 mm.. Available at AntWeb.org.

**Table 1.** Diagnostic character matrix for the new species and their most similar species in the *schulzi* group. Characters and their states are as follows. Ch01, Vertex with abundant erect setae: 0, absent; 1, present. Ch02, Pronotal humeral pair of stand setae shape: 0, Remiform; 1, filiform. Ch03, Lower mesopleuron smooth patch: 0, absent; 1, present. Ch04, Mesonotal stand setae number: 0, one pair; 1, two or three pairs. Ch05, Petiole node: 0, longer than broad; 1, broader than long. Ch06, Basigastral costulae: 0, thick, short and uniform; 1, fine, dense and very long. Ch07, first gastral tergite: 0, smooth; 1, finely striated.

| Species | Ch01 | Ch02 | Ch03 | Ch04 | Ch05 | Ch06 | Ch07 |
|---|---|---|---|---|---|---|---|
| *Strumigenys castanea* (Brown, 1953) | 0 | 0 | 0 | 0 | 1 | 0 | 0 |
| *Strumigenys itannae* **sp. nov.** | 1 | 1 | 0 | 1 | 0 | 0 | 0 |
| *Strumigenys schulzi* Emery, 1894 | 0 | 0 | 0 | 0 | 0 | 0 | 0 |
| *Strumigenys metrix* (Bolton, 2000) | 0 | 0 | 1 | 1 | 0 | 0 | 0 |
| *Strumigenys orchibia* (Brown, 1953) | 0 | 0 | 0 | 0 | 1 | 1 | 0 |
| *Strumigenys xoko* **sp. nov.** | 0 | 1 | 0 | 1 | 0 | 0 | 1 |

*Strumigenys schulzi* Emery, 1894 [27]
(Figure 4)
*Geographical Distribution*. From southern Mexico to southern Brazil.
Material examined: **BRAZIL**, **Acre**, Porto Walter, 8.270.158, −72.740.791, 19.iii.1997, on *Epipedobates* diet, (J. Cadwell) [one worker]. **Bahia**, São José da Vitória, −15.061.667, Puaias, −39.344.167, 22.v.2000, (J.R. Maia dos Santos) [one worker]. Uruçuca, Serra Grande, −14.445.833, −39.045.556, Pasto, (J.R. Maia dos Santos) [one worker], (CPDC 5651). **Ceará**, Guaramiranga, −4.266.667, −38.933.333, 28.ii.2002, ±900 m (Yves Quinet) [one worker].

−4.266.667, −38.933.333, 21.vii.2002, ±900 m (Y. Quinet) [one worker]. **Maranhão**, São Luís, −2.604.264, −44.250.581, 12.ix.2014, (J. A. Silva) [one worker, CPDC 5790]. **PANAMA**, **Colon Province**, São Lourenzo Forest, 9.240861, −79.985667, 15.x.2003, (N. Winchester and K. Jordan) [one worker]. **TRINIDAD**, **Aripo Heights**, Semi Forest litter, 14.iv.1972 (B.R. Pitkin) [1 worker] [ANTWEB-CASENT0281996].

*Diagnosis*. Basal lamella of mandible triangular, without diastema between it and masticatory margin; eyes large, with 20 or more ommatidia in total; strongly dorsoventrally compressed scapes; pronotal humeral and apicoscrobal setae short and stout hair; pilosity narrowly remiform, ground pilosity of spatulate hairs; petiole and postpetiole entirely reticulate-punctate in dorsal view, with basigastral costulae shorter than a third of postpetiole disc length.

*Comments*. The examined specimens from different localities, including the type material [28], showed variation. The main differences observed were in specimens collected in the Atlantic Forest of Southern Bahia, Brazil, in the municipality of Una, which have a significant portion of the lower mesopleuron smooth (Figure 3B), thicker and apically slightly remiform gaster pilosity (Figure 4B), differences in pilosity on the cephalic dorsum (Figure 4A) and slightly shorter petiolar node in lateral view (Figure 4B). The specimen, compared to the type, is more similar to the other examined material and has a sculptures lower mesopleuron, slightly thinner gastral pilosity, cephalic pilosity thicker (Figure 4D) and longer petiolar node in lateral view.

Although a reasonable number of specimens have been collected since Bolton's revision [4] and mainly since Brown's work [12], it remains hard to decide whether *Strumigenys schulzi* is a complex of various cryptic species or a single species that shows variation across its wide distribution. This challenge arises because, despite being a widely distributed and extensively collected species, the samples consist primarily of single individuals, making it difficult to study possible intra-populational variations.

*Strumigenys xoko* sp. nov. Santos-Neto, Chaul and Delabie

(Figures 4 and 5)

*Type material*. Holotype worker: **BRAZIL**, **Sergipe**: Canindé do São Francisco, −9.635367, −37.792324, 25.v.1997, Collected from *Aechmea aquilega* (Bispo, S.M.) [CPDC 5871, ANTWEB1048665]. Paratype queen, **BRAZIL**, **Bahia**: Iguaí, 14.643.056, −40.147.778, 04.vii.2012 (C. Leite & T. Porto et al.) [CPDC 5780, ANTWEB1048666].

*Diagnosis*. Basal lamella of mandible triangular, without diastema between it and masticatory margin; eyes large, with 20 ommatidia in total; broad flattened scapes; stand pilosity narrowly remiform, ground pilosity of spatulate setae; petiole and postpetiole entirely reticulate-punctate in dorsal view, first gastral tergite entirely thinly striated, with basigastral costulae shorter than a third of postpetiole disc length.

*Geographical Distribution*. Brazil (Bahia and Sergipe) (Figure 2)

*Description. Worker*. HW 0.43, ML 0.12, ML2 0.25, HL 0.59, SL 0.25, OL 0.08, ON 20, WL 0.53, PetL 0.26, A3+A4L 0.65, Prw 0.36, DPetW 0.16, DPetndL 0.12, PosPetW 0.25, PosPetL 0.15, GW 0.51, MtfmL 0.39, MttbL 0.3, CI 73, SI 58, MI 22, TL 2.16. (n = 1). *Head*. In full-face view, head longer than wide; occipital margin concave, meeting the vertex corners in mild indentations (Figure 4A); anterior clypeal margin slightly convex; sides gradually diverging posteriorly to the eyes. Mandible triangular, small; basal lamella subtriangular; masticatory margin with 12 teeth, the first originating without a diastemic gap between it and the basal lamella; teeth 1–5 sharp; teeth 1, 2, 4 and 5 subequal, tooth 3 largest; teeth 6 and 7 smaller than the previous; teeth 8-11 reduced denticles, tooth 12, the apical, sharp. Clypeus projecting over base of mandibles. Antennae 6-segmented. Scape broad, dorsoventrally strongly compressed; its anterior margin with three robust spatulate hairs followed by finer spoon-shaped hairs, most of them curved towards its base. Head entirely reticulate-punctate. Ground pilosity composed of subspatulate, appressed hairs; vertex with two pairs of erect setae, remiform, visibly larger than the ground pilosity of the head; apicoscrobal setae subspatulate, shorter in length to the vertex setae. *Mesosoma*. Entire mesosoma, dorsally and laterally densely reticulate-punctate, without any smooth areas

(Figure 5B). Pronotal humeral pair of setae filiform; mesonotal pair of standing setae absent; ground pilosity on dorsum of mesosoma composed sub spatulated semi-appressed setae, and three pairs of remiform. Metanotal impression forms a shallow groove. Propodeal spine is sharp, long and mostly fused to its subtended lamella, which is large and has a concave posterior margin (Figure 4B). Legs reticulate-punctate, with hairs finer than those on mesonotum and head. *Metasoma*. In lateral view, the petiole node is subquadrate, slightly higher than long; in dorsal view, wider than long; anterior margin broader than posterior, reticulate-punctate. Spongiform tissue on petiole ventrally absent, dorsally with a thin fringe on node posterior edge. Postpetiole, in lateral view, with well-developed ventral spongiform tissue; in dorsal view, dorsal edge with a few fringes, not laterally projected. Postpetiole disc reticulate-punctate. Dorsum of the metasoma covered with remiform setae. First gastral tergite entirely striated, with superimposed basigastral costulae shorter than a third of postpetiole disc length, (Figure 4C).

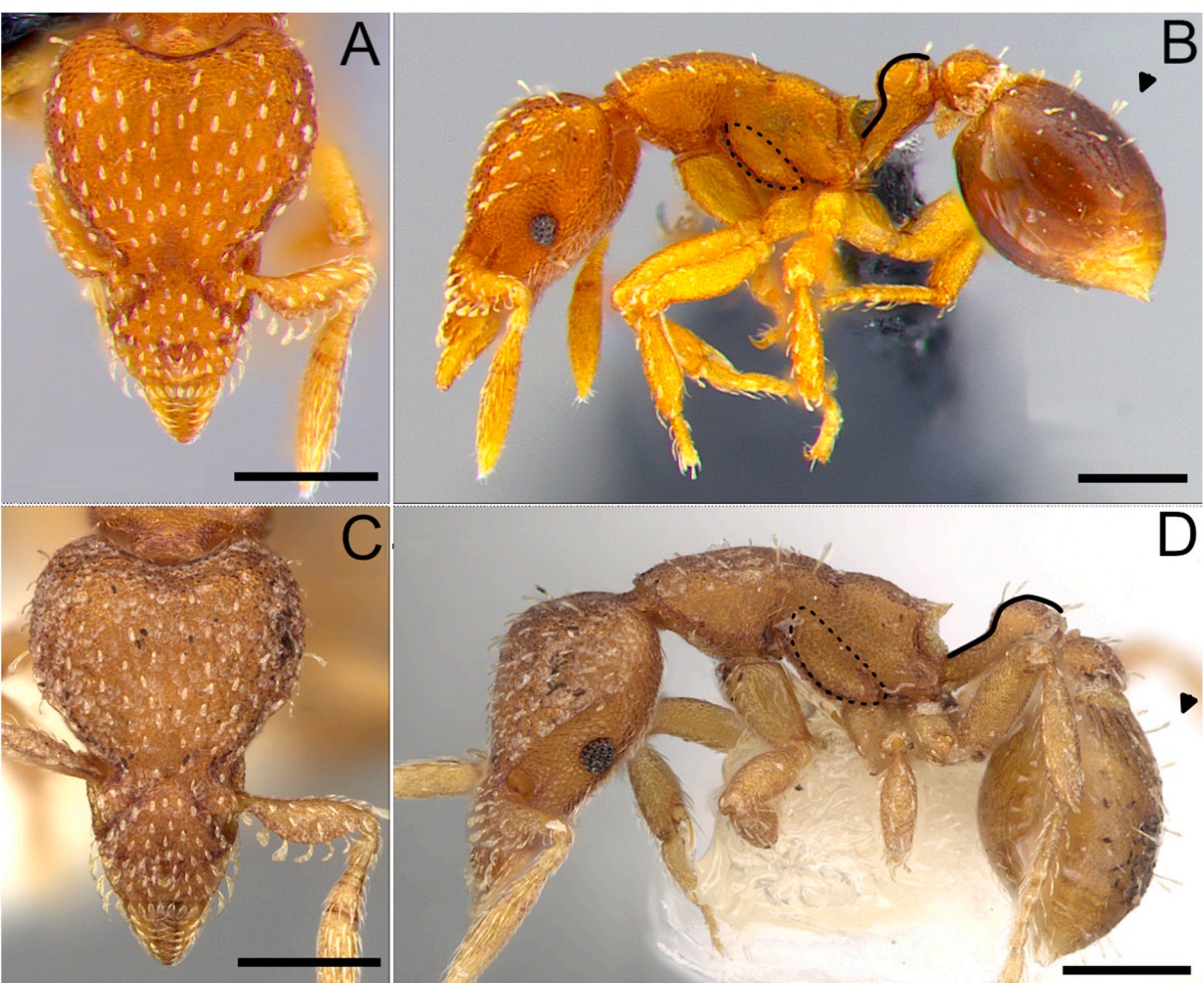

**Figure 4.** *Strumigenys schulzi* specimens; head in full face view(**A**), Lateral view (**B**)., specimen collected in Una, Bahia, Brazil. Head in full-face view (**C**). Lateral view (**D**)., specimen collected in Trinidad. [CASENT0281996]. Scale bars are 0.2 mm.. Available at AntWeb.org.

*Queen*. HW 0.49, ML 0.13, ML2 0.25, HL 0.67, SL 0.33, OL 0.15, WL 0.76, PetL 0.46, A3+A4L 0.94, Prw 0.42, DPetW 0.21, DPetndL 0.11, PosPetW 0.32, PosPetL 0.15, GW 0.63, MtfrL 0.5, MttbL 0.35. (n = 1). Similar to the worker except for differences typical of the reproductive caste: larger total length, larger compound eyes, presence of three ocelli; larger mesosoma, with small smooth patch on lower mesopleuron and numerous stand setae on mesoscutum and mesoscutellum (Figure 6).

*Etymology*. The specific epithet refers to the Brazilian indigenous Xokó people, who live on São Pedro Island in the São Francisco River, in the Brazilian state of Sergipe, close to where the type specimen was collected.

*Biology*. Little can be inferred about the biology of this ant based on the available material. The holotype was collected at the base of an *Aechmea* bromeliad, suggesting that, like other species within the *schulzi* species group, this ant prefers nesting in contact with plants.

*Comments*. The dentition pattern of *S. xoko* looks like *S. schulzi*, *S. castanea*, *S. orchibia* and *S. metrix*. *S. xoko* can be easily identified because, among similar species, it is the only one that has first gastral tergite entirely striated. (Figure 4C), (Table 1).

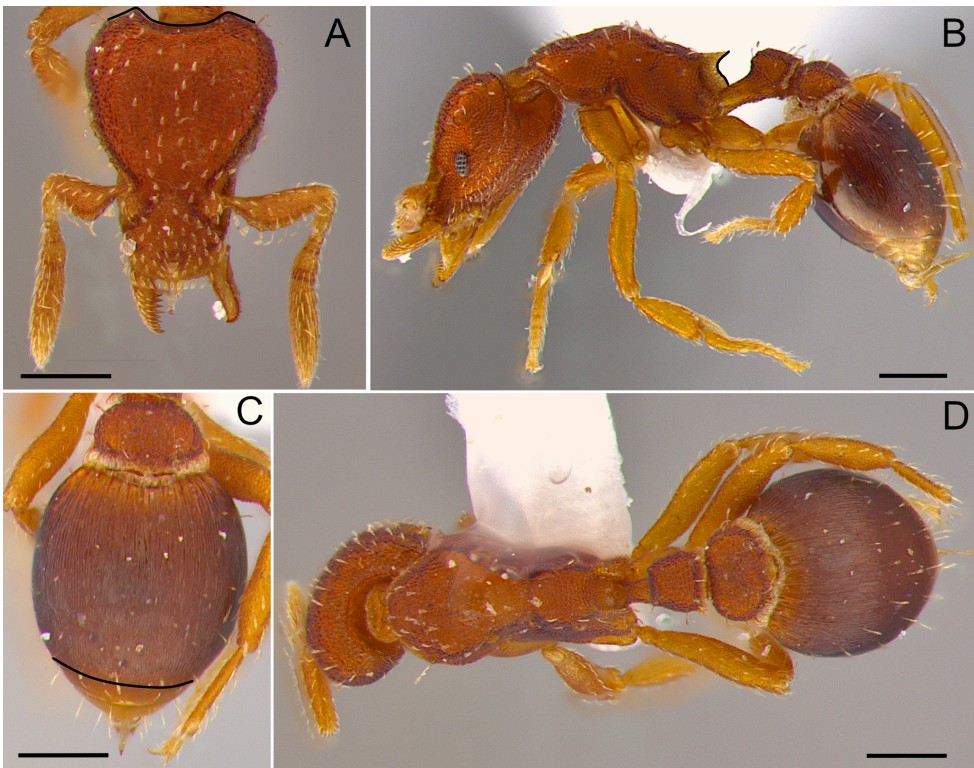

**Figure 5.** *Strumigenys xoko* sp. nov. Holotype worker, head in full-face view (**A**), lateral view (**B**), gaster in dorsal with basigastral costulae in detail (**C**), dorsal view (**D**), [CPDC 5871], [ANTWEB1048665]. Scale bars are 0.2 mm.

Identification key modified from Bolton [4]

To identify the species in the *schulzi* group described in this study, the following adaptation to the "Key to Neotropical *Pyramica* species" [4] is presented below. Starting from step 39 of the mentioned key, the following couplets were added.

**39′** Cephalic dorsum just in front of occipital margin with a transverse row of 6 small setae that are only marginally longer and more erect than the ground pilosity. Pronotal humeral setae very short, remiform. Lower mesopleuron smooth (Figure 7A) ....................
.................................................................................................................... *Strumigenys metrix*

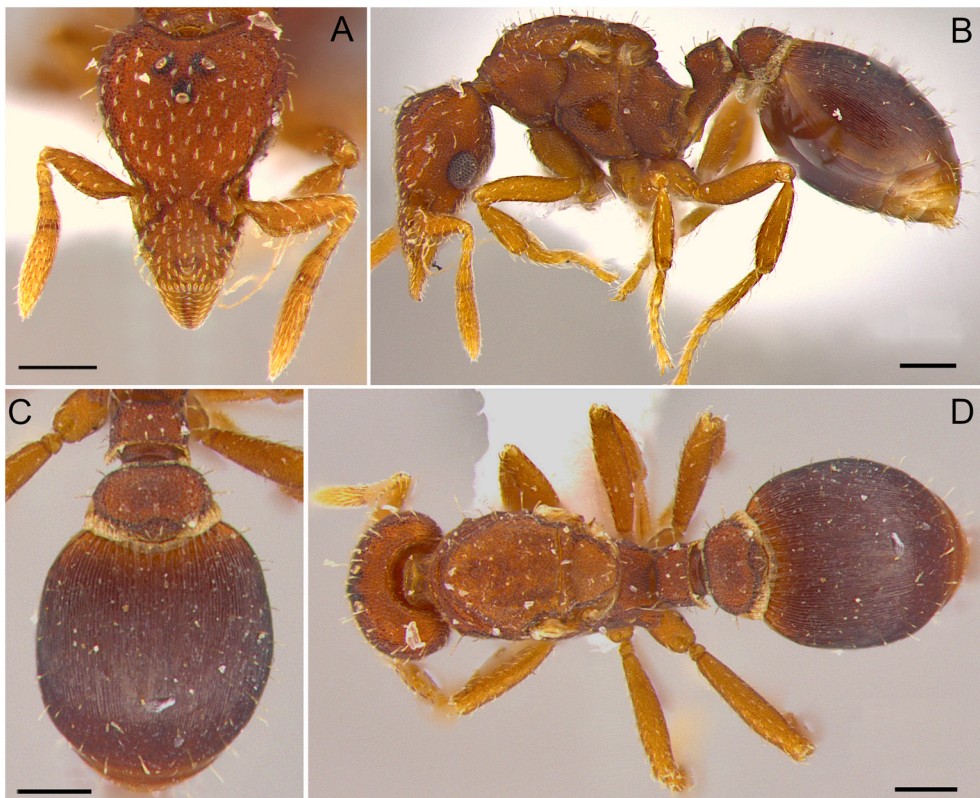

**Figure 6.** *Strumigenys xoko* sp. nov. Paratype, queen, head in full-face view (**A**), lateral view (**B**), gaster in dorsal with basigastral costulae in detail (**C**), dorsal view (**D**), [CPDC5780] [ANTWEB1048666]. Scale bars are 0.2 mm.

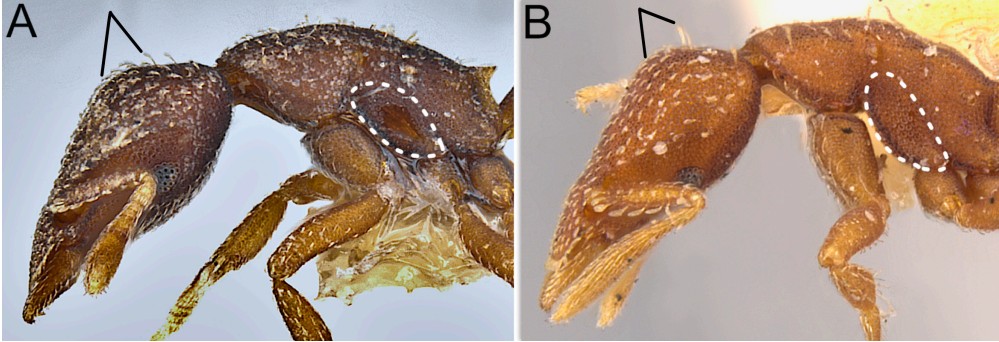

**Figure 7.** Profile view showing the difference in pilosity on the head and mesopleural area of *Strumigenys metrix* [BMNH1013599] (**A**) and *Strumigenys orchibia* [BMNH1013600] (**B**). Available at AntWeb.org.

**39″** Cephalic dorsum just in front of occipital margin with a conspicuous transverse row of 4–6 erect hairs that are obviously much longer and more erect than the ground pilosity. Pronotal humeral setae elongate, flattened or truncated apically. Lower mesopleuron entirely reticulate (Figure 7B). . . . . . . . . . . . . . . . . . . . . . . . . . . . . . . . . . . . . . . . . . . . . . .40

**40′** Vertex with abundant erect setae, without ground pilosity contrasting to the stand pilosity. Simple erect hairs on mesonotum conspicuously differentiated from the very sparse ground pilosity (Figure 8A). . . . . . . . . . . . . . . . . . . . . . . . . . . . . .*Strumigenys itannae* sp. nov.

**40″** Vertex without abundant erect setae. Erect hairs on the mesonotum and spatulate, without significant differentiation from the sparse ground pilosity (Figure 8B). . . . . . . . . . . .41

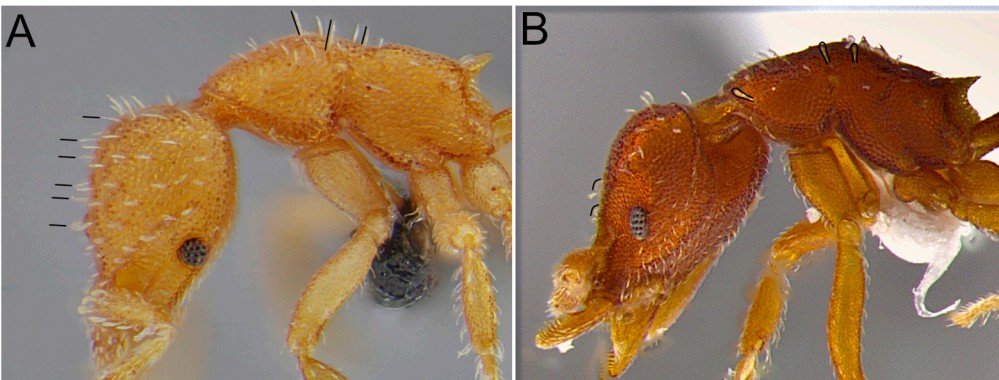

**Figure 8.** Profile view showing the difference in pilosity on the head and mesonotum of *Strumigenys itannae* sp. nov. (**A**) and *Strumigenys xoko* sp. nov. (**B**).

**41′** First gastral tergite entirely covered on thin striation, basigastral costulae very short and inconspicuous. Mesonotum with two pairs of erect setae (Figure 9A) .... … … … … … … … … … … … … … … … … … … … … … … … … … … … … … … … … … ...... … … … … … … .*Strumigenys xoko* sp. nov.

**41″** Except for basigastral costulae, first gastral tergite either entirely smooth or with thin striation on its basal half; basigastral costulae short and inconspicuous. Mesonotum with a single pair of erect setae ... 42

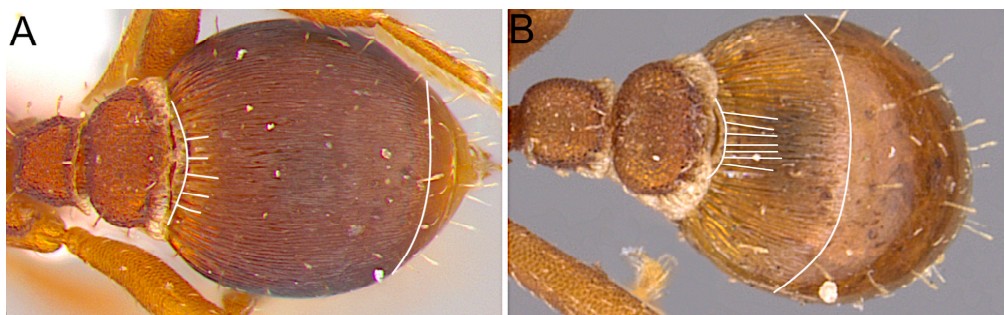

**Figure 9.** Profile view showing the difference in sculpture and basigastral costulae on first gastral tergite of *Strumigenys xoko* sp. nov. (**A**,**B**) *Strumigenys orchibia* [BMNH1013600].

42′ First gastral tergite with thin striation on its basal half, basigastral costulae adjunct and inconspicuous, merged with striation (Figure 9B). Mesonotum with a single pair of erect hairs. … … … … … … … … … … … … … … … … … … … … … … … … … ......*Strumigenys orchibia*

42″ Gaster smooth, except for basigastral costulae, which are short, sharply defined and distinctly separated. Mesonotum with 2–3 pairs of erect hairs ........................... 43

43′ Petiole node in dorsal view broader than long. … … … … … … … .*Strumigenys schulzi*

43″ Petiole node in dorsal view longer than broad. … … … … … .*Strumigenys castanea*

## 4. Discussion

*Notes on the biology of the species of the* **schulzi** *group*. Although most species of the genus are frequently associated with leaf litter and soil, repeated sampling events of different species in the *schulzi* group in epiphytes show a tendency towards arboreal habitats. For example, *Strumigenys orchibia* and *S. castanea* were described by Brown from specimens obtained in orchids of the species *Cattleya mossiae* and *C. mendelii*, intercepted in pseudobulbs from Colombia and Venezuela and kept under quarantine in the United States [12]. *Strumigenys epinotalis* is frequently recorded in bromeliads of the genus **Tillandsia** ([29,30] or in other epiphytes [4]. Recently, we recorded this species in southern Bahia in association with bromeliad on a recently fallen tree (recorded in this study).

Samples of other species in different microhabitats support the preference of some species for association with vegetation. For example, *Strumigenys schulzi*, whose types

were sampled in the vegetation (found under tree bark) [27]. A winged queen of this species was collected by E.O. Wilson in Veracruz, Mexico, also under tree bark, and other specimens were obtained by sweeping vegetation [31]. These field notes suggest, as already mentioned by Brown and Longino [2,31], that *Strumigenys schulzi* may nest in the lower strata of vegetation, using pre-existing cavities or beneath tree bark. The species was once recorded foraging on the ground in Braulio Carrillo National Park in San José, Costa Rica, suggesting it can forage on the ground and leaf litter during specific periods, such as at night [31].

Difficulty in accessing epiphytes for collections means that species associated with these plants are rarely collected. Recent collections in bromeliads in southern Bahia, Brazil [32,33] (Figure 10), have resulted in new records of species previously only known from the Amazon and other localities, such as *S. orchibia and S. metrix*. When we consider the history of more than 30 years of collections using leaf litter extraction methods and pitfall traps in this region, we find no records for these two species, and for other species, the records are quite scarce. This strongly suggests, following Brown, that these species can be associated with arboreal strata.

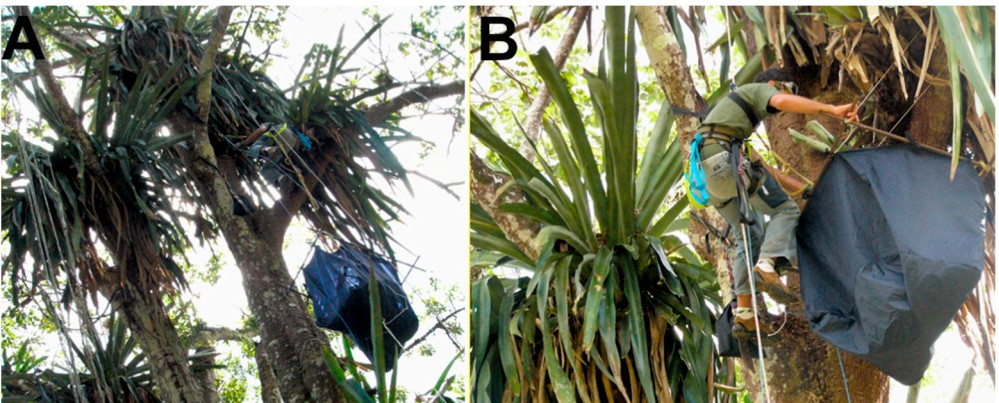

**Figure 10.** (**A**): Process of transporting the bromeliad to the ground. (**B**): sampling of bromeliads using climbing method in shading trees at Ilhéus, Bahia, Brazil. Photographs provided by Wesley DaRocha.

Records of species associated with vegetation in the genus *Strumigenys* are not limited to the *schulzi* group. Species from this group, such as *Strumigenys tococae*, Wheeler and Bequaert, 1929 [34], which nests in myrmecophyte plants of the genus *Tococae*, and *Strumigenys fairchildi* Brown, 1961 [35], also from the same group and rarely collected, with the first record for Brazil obtained from vegetation using a beating tray [36]. Another group whose species are frequently associated with plants, particularly with epiphytes, is the *precava* group (Santos-Neto et al., in prep.). All these species have morphological characteristics, such as eye size, leg size and overall size, that are typical of species with arboreal habits within the genus, as previously mentioned in [4,12,30].

**Author Contributions:** Conceptualization: E.A.d.S.-N. and J.H.C.D.; methodology: E.A.d.S.-N., J.C.M.C. and J.H.C.D.; formal analysis: E.A.d.S.-N. and J.C.M.C.; writing—original draft preparation, E.A.d.S.-N.; writing—review and editing: E.A.d.S.-N., J.C.M.C. and J.H.C.D.; supervision: J.H.C.D. All authors have read and agreed to the published version of the manuscript.

**Funding:** This research was granted by the Brazilian agencies: Fundação de Amparo à Pesquisa do Estado da Bahia (FAPESB) (EASN, N°BOL0413/2020), Coordenação de Aperfeiçoamento de Pessoal de Nível Superior (CAPES) (EASN and JCMC, Finance code 001). Conselho Nacional de Desenvolvimento Científico e Tecnológico (CNPq) (JHCD, N° CNPq 306885/2023-9).

**Data Availability Statement:** High-resolution images of the holotype of the species described here will be available on the Antweb platform (accessible at www.antweb.org) and can be accessed by

searching for the species names. Other studied specimens are available for consultation and can be searched for using the identification numbers cited throughout the text.

**Acknowledgments:** We are in debt to Ivana Cardoso for her great help with the map inserted in the paper and the revision of parts of the text. Gabriela de Figueiredo Jacintho for reading the manuscript and her suggestions, Mônica Ulysséa for sending photos for species comparison; finally, we thank the reviewers who made suggestions that greatly improved the quality of the paper.

**Conflicts of Interest:** The authors declare no conflicts of interest.

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
