# Peer review of "New Species and New Records of Strumigenys Smith, 1860 (Formicidae: Myrmicinae) from the Neotropical schulzi Species Group†"

_2673-6500, doi:10.3390/taxonomy4030032_

Round 1

Reviewer 1 Report

Comments and Suggestions for Authors  

I have no doubts about the scientific validity of the two new species and the new record species described in this manuscript. However, there are many issues with the writing of the manuscript. For example:

  1. The morphological descriptions are not concise and contain verbs, etc.
  2. The Latin species names are often written in regular font (this may have been caused by document conversion during editing).
  3. Important sections are missing, such as the lack of a "Diagnosis" for the new species.
  4. The key needs to be rewritten, for instance, a key for the schulzi species group could be created.
  5. I disagree with the author's speculation about the arboreal nature of the species.
  6. More details can be found in the document annotations.

Author Response

Comments 1: The morphological descriptions are not concise and contain verbs, etc

Response: We agree with this comment. The descriptions were indeed not concise and contained many verbs. We have revised all the species descriptions. We removed the verbs and made the writing more concise. The description of Strumigenys itannae Santos-Neto, Chaul, and Delabie sp. nov. can be found starting from line 223 and Strumigenys xoko sp. nov. Santos-Neto, Chaul, and Delabie from line 383 of the updated document.

Comments 2: The Latin species names are often written in regular font (this may have been caused by document conversion during editing).

Response: At some point, the text lost its formatting and the scientific names lost their italics. We apologize for this error. It has been corrected.

Comments 3: Important sections are missing, such as the lack of a "Diagnosis" for the new species.

Response: The diagnoses of the new species are in the document; in the first version, they begin at Diagnosis. Basal lamella of mandible triangular, without diastema between it and.. [line 81]

Diagnosis. Basal lamella of mandible triangular, without diastema between it and  [line194]

in the new document, it can be found 

Diagnosis. Basal lamella of mandible triangular, without diastema between it and [232]

Diagnosis. Basal lamella of mandible triangular, without diastema between it and [389]

We also added the diagnosis for Strumigenys schulzi, which can be found on Diagnosis. Basal lamella of mandible triangular, without diastema between it and  [line 357]

Comments 4: The key needs to be rewritten, for instance, a key for the schulzi species group could be created.

Response: We agree that the key needs to be rewritten. We have rewritten the key and added images for the steps. Regarding a key for the group, I believe that creating one would require developing a key for the species groups, which would be quite time-consuming. Adapting the steps for the currently used key is, we believe, the most feasible solution at this moment.

Comments 5: disagree with the author's speculation about the arboreal nature of the species.

Response: The group is already known to have arboreal habits; the discussion about these habits needed to be clearer. We have reformulated this section and provided a detailed discussion on the group's arboreal habits. It can be read starting from Notes on the biology of the species of the schulzi group. Although most species of the ...[line 500]  of the new document.

Thank you for your comments.

Reviewer 2 Report

Comments and Suggestions for Authors

The manuscript is worthy of publication, it would be nice if the authors embed the figures with in couplets of key. There are few typos in the text, e.g. page 5 it should be Comments not coments.

Comments on the Quality of English Language

Needs improvement, typos should be taken care-off

Author Response

Comments: The manuscript is worthy of publication, it would be nice if the authors embed the figures with in couplets of key. There are few typos in the text, e.g. page 5 it should be Comments not coments.

Response: We have rewritten the key and added images for the steps.

Thank you for your comments.

Reviewer 3 Report

Comments and Suggestions for Authors

Congratulations. Good contribution to science. I indicated some taxonomical deficiencies. 

Author Response

We corrected your comments in the document. At some point, the document lost italics formatting for the scientific names and we did not notice it; we apologize for that.

Thank you for your comments.

Reviewer 4 Report

Comments and Suggestions for Authors

Summary:

The paper entitled "New species and new records of Strumigenys Smith, 1860 (Formicidae: Myrmicinae) of the Neotropical schulzi species group" describes two new Strumigenys species and provides three new records. It further discusses morphological variations in S. schulzi and provides biological notes.

General concept comments:

Description of the new species and expanding the known distribution of the stated species groups are the strengths of the manuscript however the title and abstract lack clarity regarding which regions the records are new. The abstract may need further clarity to justify the stated objectives of the study. The manuscript lacks consistency in the use of the term schulzi-group particularly writing it in italics in some places while in regular fonts at other locations. The scientific names are also not consistently italicized reflecting the insufficient carefulness of the authors while preparing the manuscript. Use of hyphen instead of n-dash to represent range is another common problem seen in the manuscript. Further, several typos need corrections at the time of reviewing the manuscript which I did not specify here as per the reviewing guidelines.

Materials and methods need further clarity, particularly in terms of the use of terminology and body measurements. The explanation of the methods given to separate the specimens into species seems insufficient and unclear. Differential diagnosis of the newly described species lacks proper comparison with the closest relatives of the described species to diagnose them as new species. I suggest the authors provide only couplets NOT the TRIPLETS in the dichotomous key. Further, the characters provided in the key are also not contrasting. So the authors need to work further to make a clear and easy taxonomic key to Neotropical Strumigenys species. Discussion may need rewriting and reorganizing considering the results of the study. The discussion seems to provide mostly new information instead of discussing the results of the study. I assume inconsistency in in-text citations and references may not be needed to give details as a reviewer though there are a lot.

Specific comments

Line 17: clarify the new record for which locality/area

Line 18-19: not clear

Line 20: identification tool???

Line 28-30: Reference missing

Line 63-64: Reference missing

Line 64: QGIS, Reference missing

Line 66-69: Clarity needed

Line 71-72: Add more info to make it clearer

Line 86-90: Give measurement of Holotype and Paratype separately

Line 138: Start a sentence with a full generic name, write scientific names in italics (very common mistake throughout the manuscript which is not specified here for every mistake line by line)

Line 141-142: Better to be specific with comparative measurements

Line 176-177: Be specific

Line 230: Value of ML2 missing

Line 244-246: Elaborate differential diagnosis for species determination

Line 258-259: Use couplets only in dichotomous keys (see in general comments to reorganize identification keys)

Line 307: Whether it is a part of the results or discussion??

References: several errors, need corrections in most of the references

Comments on the Quality of English Language

I assume moderate editing of the English language is needed to make a manuscript standard.

Author Response

General concept comments:

Response: We added a list of new records for the group in a section of the results which can be seen starting from [line 100] of the revised manuscript.

New records Strumigenys schulzi species group [line 100]

Response: The schulzi group is a Neotropical group, and we have new records for different locations in the region, which is why we used this term in the title and abstract.

Response: At some point, the formatting of the scientific names was lost and we did not notice it; we apologize for that. We accepted the suggestions for standardizing the group name and removed the hyphens. and corrected the typos.

We inserted the list of measurements and their meanings in the section: Morphological Data Set. [line 58]

The identification key has been completely revised. The triplets were removed, and the key was transformed into a dichotomous key. Additionally, we added images for the steps of the key. [line 452]

We reformulated the discussion and added references on biological data for the species in the group. Notes on the biology of the species of the schulzi group. Although most species of the [line 500]

Specific comments

Line 17: clarify the new record for which locality/area

Response: We added a list of new records for the group in a section of the results which can be seen starting from [line 100] of the revised manuscript.

Line 18-19: not clear

The available biological data indicate that most species in the schulzi group are plant-associated species. See discussion for details Notes on the biology of the species of the schulzi group. Although most species of the [line 500]

Line 20: identification tool???

Response: Changed to identification key

Line 28-30: Reference missing

Response: reference added

Line 63-64: Reference missing

Response: reference added

Line 64: QGIS, Reference missing

Response: reference added

Line 66-69: Clarity needed

Response: Revised Species delimitation method. We compared the sets of morphological characters of the new [line 94]

Line 71-72: Add more info to make it clearer

Response: A diagnosis of the group is available in the cited reference. This part of the text was included to clarify that there was no need for another diagnosis, as the descriptions did not alter the proposal for the revision of the genus and the group.

Line 86-90: Give measurement of Holotype and Paratype separately

Response: We separated the measurements of the holotype and the paratype. [line 239]

Line 138: Start a sentence with a full generic name, write scientific names in italics (very common mistake throughout the manuscript which is not specified here for every mistake line by line)

Response: All these errors have been corrected in the text. We apologize for the lack of care.

Line 141-142: Better to be specific with comparative measurements

Response: This is observed from the shape of the lamella and the spine; morphometry is not effective for this separation. The text was revised to clarify that.  this is observed from the reduced propodeal lamella. [line 298]

Line 176-177: Be specific

Response: This part was removed from the text as it was not relevant there

Line 230: Value of ML2 missing

response: added. Queen. HW 0.49, ML 0.13, ML2 0.25, HL 0.67, SL 0.33, OL 0.15, WL 0.76, PetL 0.46 [line 427]

Line 244-246: Elaborate differential diagnosis for species determination

Response: A diagnosis of the species is provided above Diagnosis. Basal lamella of mandible triangular, without diastema between it and [line 389]. Here, we mention the difference between S. xoko and its closest relatives, which is the pattern of sculpture on the first gastral tergite in dorsal view. This character distinguishes it from all other species in the group. 

Line 258-259: Use couplets only in dichotomous keys (see in general comments to reorganize identification keys) 

Response: Key revised; see above in General concept comments

Line 307: Whether it is a part of the results or discussion??

Response: We moved this section to the results. See in the new document at line [317]

Thank you for your comments.

Round 2

Reviewer 1 Report

Comments and Suggestions for Authors

I personally believe that the revised manuscript is much improved compared to the first version. I recommend it for publication.

Author Response

Thank you very much for the revisions.

We have addressed the suggestions made by the academic editor.
No texts from Reviewer 2 to correct."